# Transport and excitations in a negative-$U$ quantum dot at the LaAlO$_3$/SrTiO$_3$ interface

Guenevere E.D.K. Prawiroatmodjo[1], Martin Leijnse[1,2], Felix Trier[1,3], Yunzhong Chen [3], Dennis V. Christensen[3], Merlin von Soosten[1,3], Nini Pryds[3] & Thomas S. Jespersen[1]

In a solid-state host, attractive electron–electron interactions can lead to the formation of local electron pairs which play an important role in the understanding of prominent phenomena such as high $T_c$ superconductivity and the pseudogap phase. Recently, evidence of a paired ground state without superconductivity was demonstrated at the level of single electrons in quantum dots at the interface of LaAlO$_3$ and SrTiO$_3$. Here, we present a detailed study of the excitation spectrum and transport processes of a gate-defined LaAlO$_3$/SrTiO$_3$ quantum dot exhibiting pairing at low temperatures. For weak tunneling, the spectrum agrees with calculations based on the Anderson model with a negative effective charging energy $U$, and exhibits an energy gap corresponding to the Zeeman energy of the magnetic pair-breaking field. In contrast, for strong coupling, low-bias conductance is enhanced with a characteristic dependence on temperature, magnetic field and chemical potential consistent with the charge Kondo effect.

[1] Center for Quantum Devices, Niels Bohr Institute, University of Copenhagen, Universitetsparken 5, 2100 Copenhagen, Denmark. [2] Division of Solid State Physics and NanoLund, Lund University, Box 118, SE-221 00 Lund, Sweden. [3] Department of Energy Conversion and Storage, Technical University of Denmark, Risø Campus, 4000 Roskilde, Denmark. Correspondence and requests for materials should be addressed to T.S.J. (email: tsand@nbi.ku.dk)

The electronic properties of many metals and semi-conductors are well described by the free-electron model or Fermi liquid theory treating electrons as effectively non-interacting. However, local interactions of electrons with bosonic modes, such as phonons, can provide attractive corrections to the Coulomb repulsion, leading ultimately to a ground state of local electron pairs[1]. The formation of pairs can be described by a negative-$U$ Anderson model, which was originally introduced to explain the electronic and magnetic properties of amorphous semiconductors[2]. This model has subsequently been applied to a wide range of phenomena, such as the transport of atoms in optical lattices with attractive interactions[3], and plays an important role in the theory of pairing without a global superconducting phase, relevant to unconventional and high $T_c$ superconductors[1]. Also, the widely debated pseudogap phase observed above the transition temperature[4] may originate from a phase of preformed local electron pairs that exist above the superconducting critical temperature $T_c$. Such preformed pairs differ from the Cooper pairs in conventional BCS (Bardeen–Cooper–Schrieffer) superconductors, which form in momentum space, simultaneously with the superconducting phase transition[1].

Despite the general importance of attractive interactions and local pair formation, only very recently has paring been demonstrated at the level of single electrons in a single-electron transistor at the LaAlO₃/SrTiO₃ (LAO/STO) interface[5–7]. Direct evidence was found for pairing at magnetic fields and temperatures significantly beyond the critical values of superconductivity. The negative-$U$ scenario modifies the allowed transport processes and generates an excitation spectrum qualitatively different from the case of conventional quantum dots. At low bias voltage, current flows by second-order pair tunneling

and thermally excited sequential single-electron tunneling and for weak tunnel coupling and low temperature transport is highly suppressed[8]. In the regime of strong tunnel coupling, electron pairing is predicted to suppress the spin Kondo effect, one of the most well-known many-body phenomena in conventional quantum dots (QDs)[9–15]. Instead, however, a many-body charge Kondo effect has been predicted, where higher-order cotunneling of electron pairs effectively establishes a Kondo resonance at the degeneracy points of even charge states[16, 17]. The charge Kondo effect has been theoretically predicted to play an important role for superconductivity[18, 19] in materials with negative-$U$ impurities.

Here we study the low-temperature transport properties of a negative-$U$ QD defined at the interface of LAO/STO by local electrostatic gates allowing tuning of the charge occupation and tunnel couplings. For weak coupling we perform transport spectroscopy of the excitations of the paired ground state as a function of voltage bias, chemical potential and magnetic field. The excitation spectrum and the appearance of energy gap at low bias voltage is qualitatively different from the situation in conventional QDs and in good agreement with calculations based on a perturbation theory approach to the Anderson model with a negative-$U$. For strong coupling, the gap is replaced by an enhanced contribution from pair tunneling at low temperature with a dependence on magnetic field, chemical potential and temperature consistent with the predictions for the charge Kondo effect expected in this regime[16, 17].

## Results

**Paired ground state in a gate-defined LAO/STO QD.** Our device was fabricated at the LAO/STO interface by conventional lithographic techniques. An LAO top film was deposited by

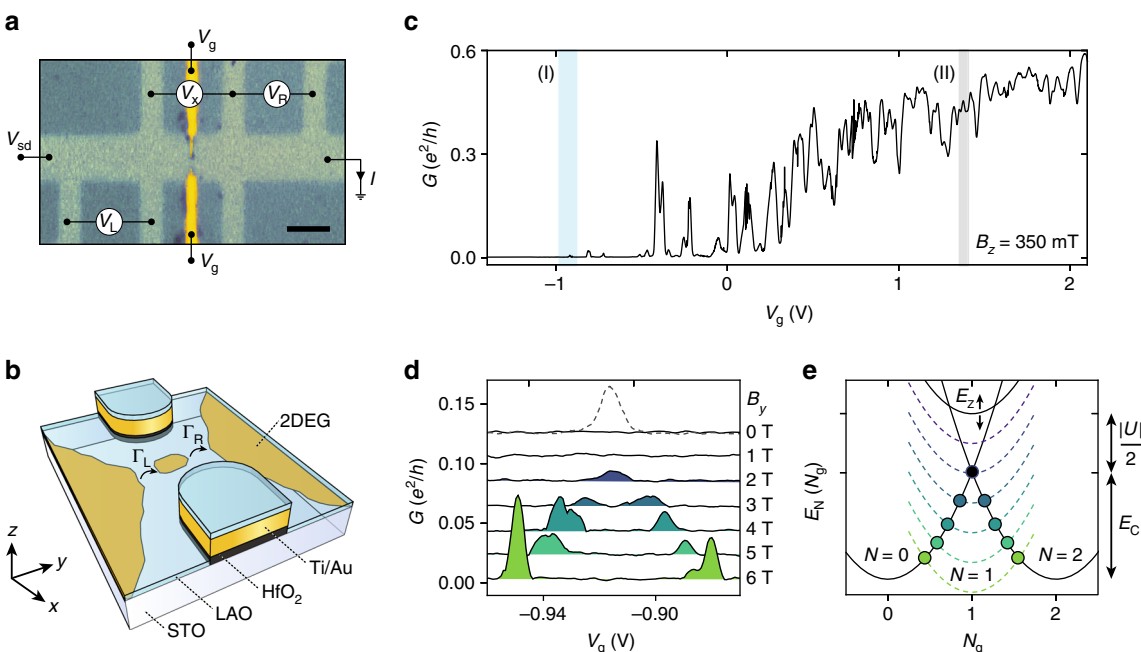

**Fig. 1** Device and zero-bias transport characteristics. **a** Optical microscope image of the split-gate device. The *dark color* shows the LSM hard mask and the STO is exposed to the LAO at the *light colored regions* forming a two-dimensional electron gas (2DEG). *Scale bar* is 5 µm. The 200 nm wide metal top gates are 1 µm apart. **b** Schematic of the device and formation of the QD close to pinch off with tunnel couplings $\Gamma_{L,R}$ to the 2DEG. **c** Conductance as a function of $V_g$ showing depletion for $V_g < -1$ V. The two regions I and II, of weak and strong tunnel coupling, respectively, are investigated in detail. **d** Conductance $G$ ($V_g$) for $V_{sd}=0$ in region I with varying magnetic field $B_y=0$ to 6 T (*solid lines*). For clarity, traces are vertically offset by ~0.012 $e^2/h$. For $B=0$ a single peak is observed when increasing the bias to $V_{sd}=160$ µeV (*dashed*). **e** Dependence of the ground-state energy on gate-induced charge $N_g=C_gV_g/2e$ for a single orbital with occupation $N=0,1,2$ and effective negative charging energy $U$. *Solid lines* represents $B_y=0$ and *dashed lines* illustrate the Zeeman splitting of the odd-$N$ state at $B_y=1$–6 T. For $E_Z \geq |U|$, sequential single electron transport is allowed at the points marked with *circles*

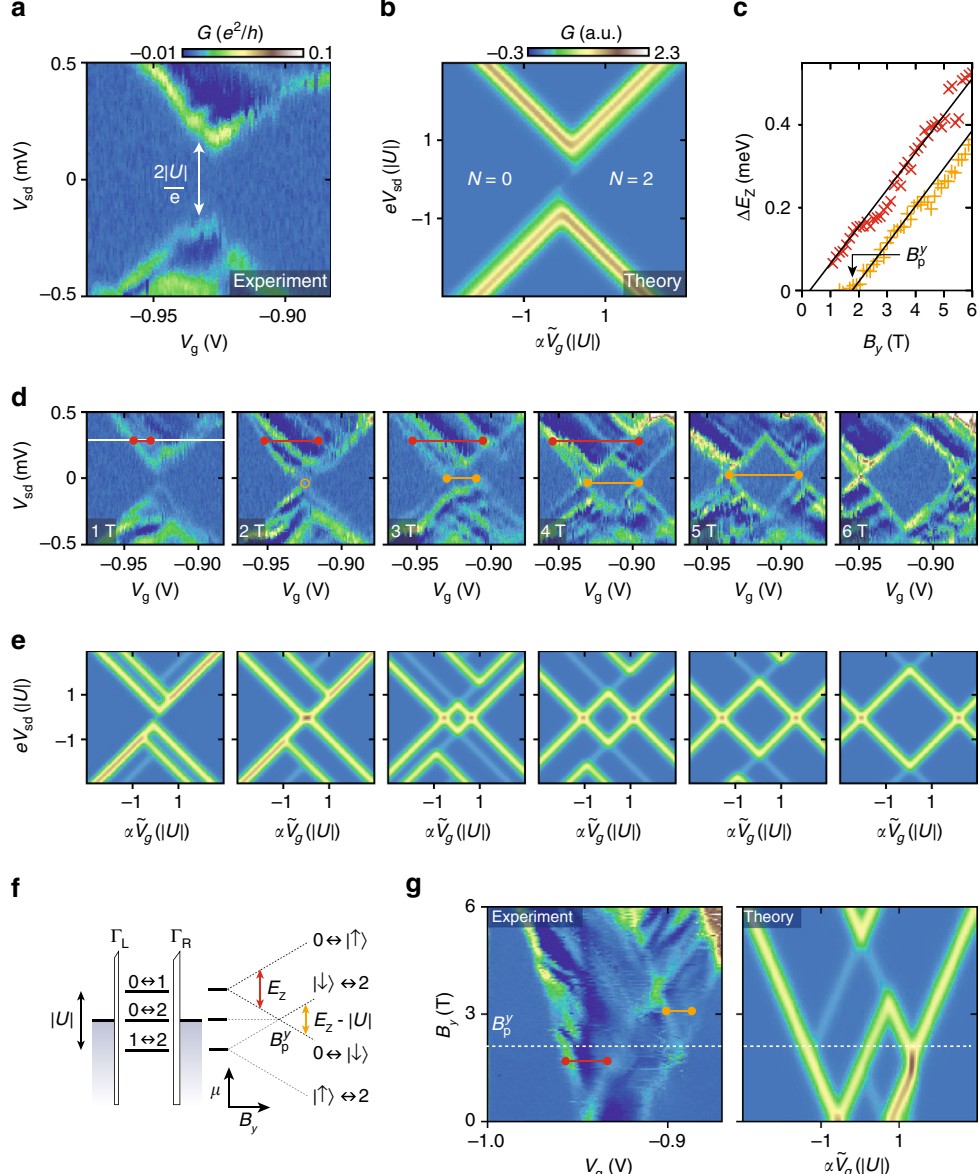

**Fig. 2** Excited state spectroscopy. **a** Bias spectroscopy in the region of a single-pair transition shown in Fig. 1d. The transport gap $|U|$=160 μeV is indicated. **b** Corresponding transport calculation of the bias spectroscopy based on a single-orbital Anderson model with attractive interactions and occupation $N$. **c** Zeeman energy splitting $\Delta E_Z$ extracted from measurements in **d**, **g**. The energies for the ground (excited) state transitions are indicated by *orange* (*red*) markers. The linear fits yields $|g|$=1.5 and intercept $B_p^y$ = 1.8 T. **d** Evolution of the bias spectroscopy measurements with magnetic field $B_y$ with corresponding calculations shown in **e**. The color scales for **d**, **e** are as in **a**, **b**, respectively. **f** Schematic illustration of the electrochemical potentials for the relevant transitions in the QD and their evolution with magnetic field. For the splitting of the ground state, $\Delta E_Z=E_Z-|U|$ (*orange line*), and for the splitting of the excited state (*red line*) $\Delta E_Z=E_Z$, where $E_Z=g\mu_B B_y$ is the Zeeman energy. The associated features are indicated in the bias spectra (**d**, **g**) by *orange/red* lines. **g** Measured and simulated magnetic field dependence of $G(V_g)$ at $V_{sd}$=300 μeV (*white line* in **b**), highlighting the magnetic field dependence of the excited state spectrum

room-temperature pulsed laser deposition on STO substrates[20], prepared with a Hall bar mask pattern[21] combined with nanoscale electrostatic topgates, arranged in a split-gate geometry[22–27] (see Methods for details). Figure 1a, b shows a micrograph and schematic of the device. The two-dimensional electron gas (2DEG) on either side of the gates was characterized separately and exhibits a typical sheet resistance $R_s$~ 600 Ω/sq at $T$=1 K and a superconducting phase for temperatures below 270 mK with critical magnetic fields $B_c^z \sim$ 90 mT and $B_c^y \sim$ 2 T for the out-of-plane and in-plane directions, respectively[28] (details are presented in Supplementary Note 1).

The differential conductance $G = \frac{dI}{dV_x}$ as a function of gate voltage $V_g$ applied simultaneously to both gates is shown in

Fig. 1c, where superconductivity in the leads adjacent to the QD is suppressed by a magnetic field $B_z$=350 mT. The split-gate geometry creates an electrostatic saddle potential locally depleting the 2DEG and for $V_g \lesssim -1$ V the conductance is pinched off. For $V_g \sim -0.9$ V (region I in Fig. 1c) narrow conductance peaks separated by regions of no conductance are found. For higher $V_g$, the peak density increases and peaks are significantly broader. Such behavior is common in split-gate devices[22, 24, 29, 30], where disorder-induced fluctuations in the potential landscape result in tunneling through a localized conducting region, i.e., a QD (Fig. 1b), rather than idealized one-dimensional channels. Indeed, disorder is commonly observed for LAO/STO heterostructures[20, 28, 31, 32] and QDs have recently been reported for LAO/STO split-gate devices[22]. For $V_g$ close to pinch-off our

simple geometry thus realizes a QD in which the occupation $N$, size and tunnel couplings $\Gamma_{L,R}$ are simultaneously tuned by $V_g$. While multiple dots could in principle be formed in the junction the device is dominated by a single QD as shown below. In the following we first focus on the regime of weak tunnel coupling ($V_g \sim -0.9$ V; region I in Fig. 1c) and then proceed to consider a regime with stronger tunnel coupling ($V_g \sim 1.38$ V; region II in Fig. 1c).

To investigate the occupation of the ground state, Fig. 1d shows $G(V_g)$ in region I at low bias voltage $V_{sd}$ for increasing magnetic fields $B_y = 0$ to 6 T at $B_z = 0$ T. The zero-bias conductance is featureless for $B_y = 0$ and 1 T; however, at $B_y = 2$ T a single peak appears that splits linearly into two peaks upon further increasing $B_y$. When increasing the bias voltage to $V_{sd} = 160$ µV, a conductance peak is also observed at $B = 0$ (dotted trace in Fig. 1d).

In agreement with ref. [5], this ground state behavior is consistent with attractive electron–electron interactions that reduce the energy of the even-$N$ charge states by an amount that exceeds the electrostatic energy $E_C = e^2/C_\Sigma$, required for charging the QD by one electron. Here, $C_\Sigma$ is the total capacitance of the QD. For a single-energy level this is equivalent to an effective negative charging energy $U$[8] which favors double occupation and constitute an effective pair binding energy. The $N$-dependent part of the ground-state energies can then be effectively described by $E_N(N_g) = E_C(N - N_g)^2 - \tilde{p}_N(E_C + |U|/2)$. Here, the first term accounts for the conventional electrostatic contribution, and the second term for the favoring of even-$N$ states. $N_g = C_g V_g/2e$ is the gate-dependent excess charge on the dot[33] and $\tilde{p}_N = 1, 0$ for $N$ even and odd, respectively. The result is shown in Fig. 1e. At zero $B_y$, no degeneracy points of even- and odd-$N$ ground states appear, thus preventing single-electron sequential tunneling. At the degeneracy points between even-$N$ ground states ($N = 0, 2$) (black circle in Fig. 1e), low-bias transport of electron pairs occurs by second-order tunneling, which is highly suppressed for weak tunnel coupling. Sequential single-electron tunneling is, however, accessible at a finite bias voltage $V_{sd} \geq |U|$, consistent with the results in Fig. 1d. For increasing $B_y$, the spin-degenerate odd-$N$ state splits by the Zeeman energy $E_Z = g\mu_B B_y$ ($g$ is the Landé $g$-factor and $\mu_B$ the Bohr magneton) allowing first-order single-electron tunneling at $V_{sd} = 0$ for $E_Z \geq |U|$ as an odd-$N$ state move below the pair degeneracy point (blue-green markers in Fig. 1e). In this case, two even–odd degeneracies ($0 \leftrightarrow 1; 1 \leftrightarrow 2$) appear with a spacing $\Delta V_g$ that linearly increases with $B_y$, in agreement with the experiment. We note that since $V_g$ tunes both density and tunnel couplings, the current level drops below the detection limit before the dot is depleted and the absolute occupation of the dot cannot be assigned from these measurements. In the following we focus on the properties of a single orbital occupied by $N = 0, 1$ or 2 electrons.

**Excited state spectrum of the negative-$U$ QD.** Further insight into the properties of the negative-$U$ QD can be gained from the excited state spectrum. At finite bias, $G(V_g)$ provides a spectroscopic probe of the level structure of the QD as shown in Fig. 2a, d, for the $V_g$ range of the pair transition studied in Fig. 1d. Subsequent resonances show similar behavior (Supplementary Note 2). The bias spectroscopy reveals diamond-shaped regions of low conductance, characteristic for transport through a QD. However, in contrast to conventional QDs, for $B = 0$, diamonds do not close but exhibit a 'pairing' gap $|V_{sd}| < |U| = 160$ µeV, which decreases with $B_y$ (Fig. 2d), and has closed at $B_y = 2$ T[6] corresponding to the situation where zero-bias conductance appears in Fig. 1d. For fields above $B_y = 2$ T, a new diamond emerges with a width that increases linearly with $B_y$ (orange line in Fig. 2d).

At high $V_{sd}$, a discrete excitation spectrum is clearly observed as lines parallel to the diamond edges. We note that this is in

contrast to transport in metallic superconducting islands which may exhibit similar ground-state behavior[33] but display a continuous density of states above the gap. The continuous evolution of the excited states in a magnetic field is highlighted in Fig. 2g, which shows $G(V_g)$ at finite $V_{sd} = 300$ µV (white line in Fig. 2d) as a function of $B_y$. The range of finite conductance widens linearly with $B_y$, consistent with the closing of the pairing gap and subsequent splitting of the diamonds in Fig. 2a, d. A linear Zeeman splitting of the first transition is clearly observed (red line), and at the field at which the ground-state transition takes place, a new excitation emerges associated with the $N = 1$ diamond (orange line). In Fig. 2c both splittings have been extracted and converted into energy $\Delta E_Z$ using the gate lever arm $\alpha = 0.005$ estimated from the slope of the diamonds (Supplementary Note 2). A linear fit $\Delta E_Z \propto g\mu_B B_y$ yields a $g$-factor of 1.5 and extrapolating $\Delta E_Z(B_y)$ of the ground-state splitting (orange markers) to zero provides an accurate estimate of the pair-breaking field $B_P^y = 1.8$ T[5]. This gives a Zeeman energy $E_Z \approx 160$ µeV = $|U|$, confirming that the pair-breaking field indeed relates directly to the pairing energy $|U|$.

We modeled the system as a single-orbital Anderson model with an effective negative-$U$[8]. The model is the single-orbital version of the Hubbard model proposed in ref. [5], tunnel-coupled to a Fermi sea and allowing calculation of the transport currents at finite bias. Figure 2b, e shows the results of transport calculations based on complete next-to-leading order perturbation theory in the tunneling between the QD and the leads, including all coherent one- and two-electron tunneling processes, such as single-electron tunneling, cotunneling and pair tunneling[34, 35]. To match the experiment we take temperature $T = 23$ mK and the parameters $|U| = 160$ µeV, $g = 1.5$. Asymmetry of the barriers results in a different visibility of the conductance peaks associated with the alignment of the QD levels with the chemical potential of the two reservoirs and coupling strengths $\Gamma_L = 10\Gamma_R = k_B T/10$ ($k_B$ is Boltzmanns constant), was chosen to match the asymmetries in Fig. 2a, d. The perturbation model has been shown to accurately describe transport in conventional QD[34, 35], and the only difference here is that we take $U$-negative. Details are presented in Supplementary Note 4. The agreement between experiment (Fig. 2a, d) and transport calculations (Fig. 2b, e) is excellent, including the ground-state evolution and the excitation spectra. The bias spectroscopy can be understood by considering the chemical potentials shown in Fig. 2f, for adding electrons to the single orbital with a negative charging energy for the double occupied state. The transitions corresponding to the two visible excitations in Fig. 2d, e have been indicated. While the $N = 0 \leftrightarrow 2$ ground-state transition has no field dependence below the pairing field, the transitions including odd-$N$ excited states exhibit a linear Zeeman splitting with $B_y$.

We note that from comparison with theory calculations, it is evident that the measured spectrum contains a number of excitation lines that are not predicted by the model. A possible origin are resonances in the low-dimensional leads connecting to the QD, consistent with the negative differential conductance observed upon tuning the states off resonance (e.g., dark blue areas in Fig. 2d). In the weak coupling regime no features associated with the superconducting transition of the leads have been observed.

**Temperature dependence for weak tunnel coupling.** At the $N = 0, 2$ degeneracy point, low-bias conductance at $B = 0$ results from a combination of second-order pair tunneling and thermally excited sequential tunneling. This is expected to lead to a characteristic temperature dependence, different from conventional QDs where transport is dominated by sequential tunneling[8]. Figure 3a, e shows the temperature dependence of $G(V_g)$ for $B_y = 0$ and $B_y = 6$ T,

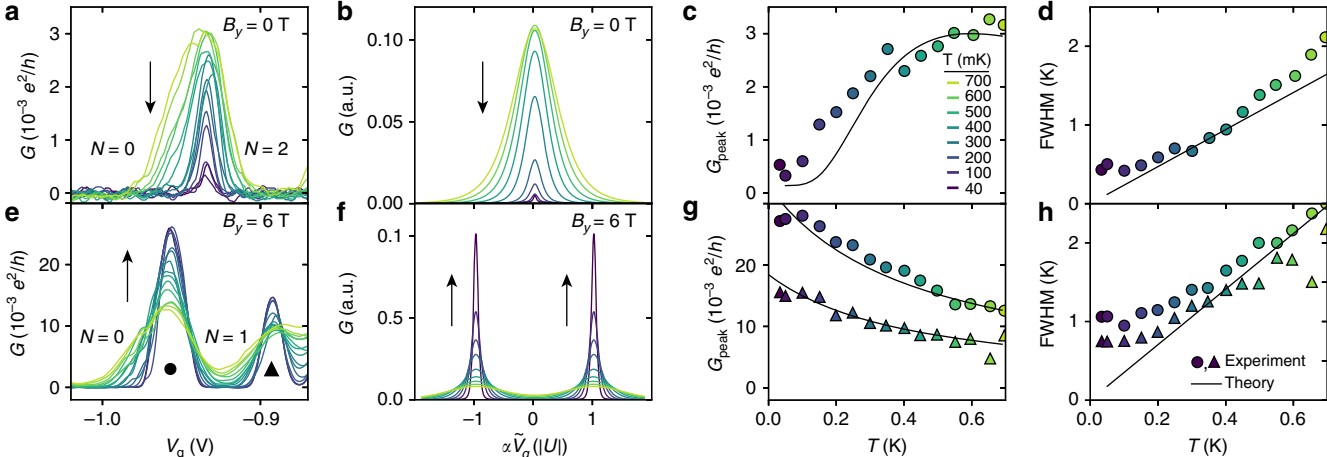

**Fig. 3** Temperature dependence of zero-bias conductance. **a** $G(V_g)$ for varying temperature $T$ at $B_y$=0 T. $N$ labels the occupation of the orbital, and the color coding of the different temperatures relevant for all panels is shown in **c**. **b** Corresponding theory calculations based on the negative-$U$ Anderson model. *Arrows* indicate the qualitative temperature dependence of the peak conductance. **c** The temperature dependence of the fitted conductance peak amplitude $G_{peak}$ and **d** full width at half maximum (FWHM). The results of the simulations are shown as *solid lines*. **e–h** Shown are the results corresponding to **a–d** measured with $B_y$=6 T, where transport is dominated by conventional sequential tunneling. The resonance peak has split into two peaks indicated by markers ● and ▲

respectively, with the corresponding calculations presented in Fig. 3b, f (Supplementary Note 5). The extracted peak heights and widths for the two cases are presented in Fig. 3c, d, g, h. The measurements allow a direct comparison of the negative-$U$ situation ($B_y$=0) and the sequential tunneling case ($B_y$=6 T). Considering first the peak heights, it is clear from Fig. 3a–d that two opposite trends are observed. For $B_y$=6 T, the peak height increases with decreasing $T$ as $\propto 1/T$, as expected for sequential tunneling in the limit $\Gamma \ll k_B T \ll \Delta E, E_C$[36]. The result is consistent with our calculations and confirms the discrete spectrum of the QD. For the $B$=0 case, the combination of thermally excited single-electron transport and second-order pair tunneling leads to a completely different non-monotonic decrease in peak height (Fig. 3c) which is in good agreement with the experimental results.The corresponding peak widths shown in Fig. 3d, h exhibit thermal broadening in both cases in agreement with the calculations. The saturation of the width at low temperature is assigned to a lifetime broadening.

**Enhancement of pair tunneling for strong coupling**. Having analyzed the weak coupling regime, we now consider a regime with stronger coupling at $V_g \sim 1.3$ V (region II, Fig. 1c). Figure 4a shows the bias spectra in a region containing several resonances. The increased coupling leads to a broadening of the conductance features, but the diamond-shaped pattern remains evident. Compared to the regime of weak coupling, the overall diamond size (set by level spacing, charging energy and $|U|$) has reduced to $\sim$100–200 µeV, consistent with a larger effective size of the QD. Surprisingly, no transport gap is observed and diamonds merge at $V_{sd}$=0 meV in broad, high-conductance peaks. In the following we focus on a single such resonance (arrow in Fig. 4a); however, the other resonances show similar characteristics (Supplementary Note 6). Upon increasing $B_y$, Fig. 4b shows a clear splitting of the resonance peak at $B_p^y \sim 1$ T confirming a paired ground state. We consider first the simpler case where a perpendicular field $B_z$=300–350 mT was applied to suppress superconductivity in the leads ($B_c^z \sim 90$ mT). The in-plane magnetic field and temperature dependencies are shown in Fig. 4c, e. Compared to the regime of weak coupling two striking differences are observed. First, the height of the conductance peak $G_{peak}$ increases with decreasing the temperature below $\sim$250 mK (Fig. 4g) and, second, the height of the conductance peak in the electron pairing regime for $B_y$=0 T

exceeds the conductance in the sequential tunneling regime for $B_y > B_p^y$ (Fig. 4c). Both trends are qualitatively opposite of what was observed for weak coupling (Figs 1d, 3a, d), and cannot be described by the perturbative transport calculations used above. As we will show below, these observations are consistent with the so-called charge Kondo effect.

In conventional QDs in semiconductors and molecules, a well-known many-body phenomenon that occurs in the regime of strong tunnel coupling is the spin Kondo effect[9–15]. The effect appears for odd QD occupation where coherent higher-order cotunneling processes effectively screen the unpaired spin on the QD, leading to a many-body Kondo resonance at the Fermi level. A key signature of the spin Kondo effect[37] is an increase of conductance in the odd-$N$ Coulomb valley, for decreasing $T$ around the characteristic Kondo temperature $T_K$ determined by $U$ and the tunnel coupling $\Gamma$. For a QD with negative-$U$, such as the present case, the spin Kondo effect is prevented as the even-$N$ ground states do not support unpaired spins. Instead, however, a charge Kondo effect has been predicted[17] at the degeneracy point of even charge states, e.g., $N$=0,2. At these points, a coherent superposition of pair cotunnel processes that use virtual, oddly occupied intermediate states effectively screen the occupation number on the QD and can establish a charge Kondo resonance at the Fermi level. Interestingly, the situation can be directly mapped to the conventional spin Kondo model. Here, the 0 ↔ 2 charge degeneracy takes the role of a pseudo-spin, and the roles of the gate voltage and the magnetic field are interchanged[16]. At the 0 ↔ 2 charge-degeneracy point, the charge Kondo effect then lifts the transport blockade observed for weak coupling and, in agreement with the experiment, generates a conductance resonance which increases upon decreasing $T$ below the Kondo temperature $T_K = \sqrt{2U\Gamma}/\pi \exp(-\pi U/8\Gamma)$ determined by $U$ and $\Gamma$[16].

In analogy with the spin Kondo effect, the conductance increase of the charge Kondo resonance is expected to follow $1/\log^2(T/T_K)$ for temperatures above $T_K$ and saturate for $T \ll T_K$ to $2 e^2/h \times 4\Gamma_L\Gamma_R/(\Gamma_L+\Gamma_R)^2$. As shown in Fig. 4g, the height of the resonance peak does not saturate, showing that $T_K$<250 mK and possibly smaller than the electron temperature of the experiment. In this regime, the conductance increase of the Kondo effect has been shown to follow $-\log(T)$[38], which indeed provides a good fit to our data over an order of magnitude in $T$ (Fig. 4). This is in reasonable agreement with $T_K$ estimated from

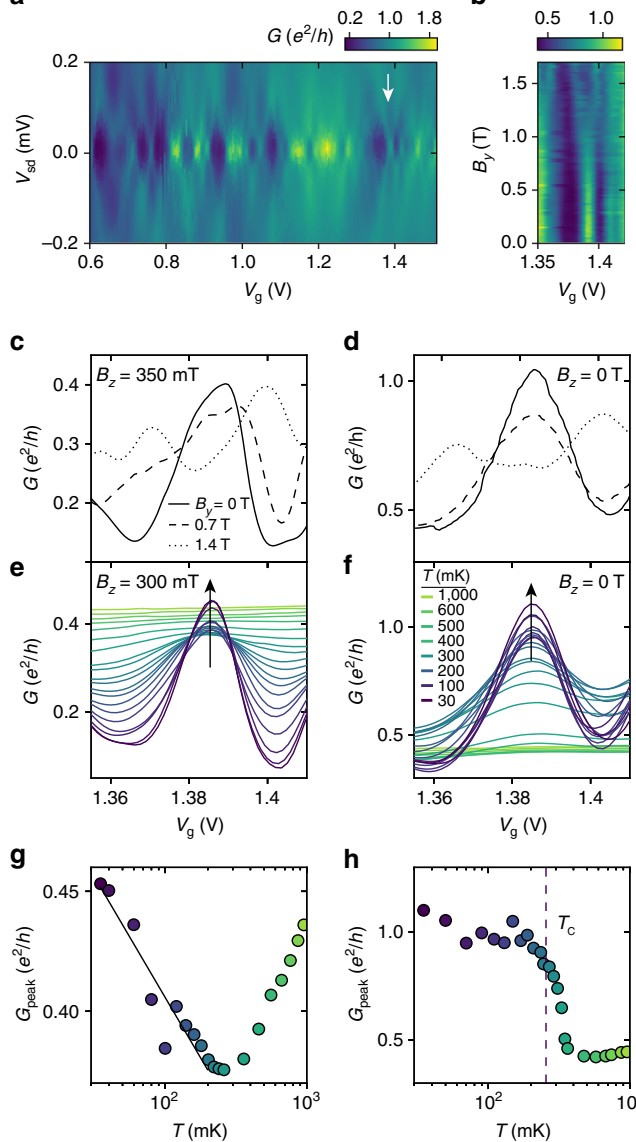

**Fig. 4** Transport characteristics of the strong tunneling regime. **a** Bias spectroscopy of the conductance $G(V_g)$ in the strong coupling regime $V_g > 0.6$ V for $B = 0$, showing diamond-shaped regions of suppressed conductance. The zero-bias resonance at the gate voltage indicated by the *white arrow* is studied for varying in-plane magnetic field $B_y$ **b–d** and temperature **e**, **f** with and without out-of-plane magnetic field $B_z > B_c^z$ to suppress superconductivity in the leads. In **e**, **f** the *arrows* emphasize the temperature dependence of the peak conductance. **g** Extracted peak height from **f**, fitted to $-\log(T)$ (*black line*). **h** Extracted peak height from **e** with $T_c$ of the leads (*dashed line*) found from independent measurements of the leads

the width in the resonance peak, full width at half maximum $= 4k_B T_K$[39], which gives $T_K \sim 400$ mK for the widths in both $V_g$ and $V_{sd}$ (Supplementary Note 5). We note that both the temperature dependence and width may be influenced by the finite $B_z$ or contributions from temperature-assisted sequential tunneling which is not taken into account. Further, the conductance increase for $T < 250$ mK could be explained by small regions of superconductivity remaining in the part of the leads making contact to the QD, despite the applied $B_z$. However, such a scenario seems unlikely, as it would require a critical temperature of $\sim 250$ mK at $B_z = 300$ mT. The values reported so far of the critical fields for both bulk and nanoscale LAO/STO devices do not exceed 300 mT[28].

A distinct difference between the spin Kondo effect and the charge Kondo effect is their relation to superconductivity in the leads. The spin Kondo effect competes with superconductivity[40] as Cooper pairs are incapable of screening the unpaired spin. On the other hand, the coherent tunneling of electron pairs leading to the charge Kondo effect is compatible with superconductivity[18, 19] and can even enhance the probability for paired electrons in the surrounding leads and has been predicted to enhance $T_c$, or even in some cases, act as pairing mechanism for superconductivity[18, 19]. This cooperation of the two phenomena is consistent with the result of the experiment in Fig. 4d, f, h which shows the corresponding measurements for $B_z = 0$ T where the leads are superconducting. The zero-bias peak conductance is strongly enhanced below 350 mK close to the transition temperature $T_c = 270$ mK measured independently for the leads.

The original theory of the charge Kondo effect was developed by Taraphder and Coleman[17], and was used to explain the low-temperature resistance increase observed in bulk samples of the small-gap semiconductor PbTe doped with Tl which acts as a negative-$U$ center[41, 42]. We note that the charge Kondo effect in the negative-$U$ QD is different from the two-channel charge Kondo effect studied by Iftikhar et al.[43]. To the best of our knowledge, the present experiment is the first indication of the pair charge Kondo effect in a tunable system where the presence of attractive interactions that result in electron pairing can be independently verified.

## Conclusion

While we have presented detailed measurements of two resonances, the remaining resonances show similar characteristics (Supplementary Notes 2, 3, 5 and 6). The magnitude of $U$ fluctuates between resonances with an overall decrease upon increasing $V_g$[5]. Similar characteristics were reproduced in subsequent cool-downs of the device. Various mechanisms can lead to attractive electron–electron interactions such as coupling to phonons, plasmons, excitons, valence-skipping defects and dopants, etc. (see ref. [1] and references herein). Our experiment does not reveal the origin of the attractive interaction in STO; however, the results demonstrate that local pair formation is a general property of LAO/STO devices and hence not unique to devices created by the atomic-force microscopy sketching method[5]. Further, our results support the validity of the physical interpretation first presented by Cheng et al.[5]. The standard lithography-based fabrication scheme employed in this study allows for future investigations of the relationship between the negative-$U$ and specific STO doping mechanisms and our work highlights the potential of mesoscopic complex oxide devices, both for exposing the unconventional properties of complex oxide interfaces and for studying mesoscopic transport phenomena in parameter regimes of fundamental interest. In the future, oxide devices with added complexity and individual control of charge occupation and tunnel coupling may provide further understanding of the charge Kondo effect and the relation between preformed electron pairs and the superconducting phase.

## Methods

**Sample fabrication.** A hard mask was created by growing a layer of LaSrMnO$_3$ (LSMO) by pulsed laser deposition (PLD) at room temperature on the STO substrate. Nanoscale topgates were defined by standard lithography. A 30 nm layer of HfO$_2$ was grown by atomic layer deposition at 90 °C to serve as a barrier for leakage currents to the interface. A stack of Ti/Au was deposited on top by standard metal evaporation techniques. The gates are 200 nm wide and have a lateral separation of ~1 μm. After defining the topgates, the LSMO hard mask was patterned by electron-beam lithography and wet etched into a Hall bar pattern. Finally, a 16 nm top layer of LAO was grown with PLD at room temperature to create the conducting interface. Details of the hard mask growth and wet etch are described in ref. [21]; we note that no signatures of possible magnetism in the amorphous LSMO top layer has been observed in any of the measurements.

**Low-temperature measurements**. Measurements were performed in a dilution refrigerator with a base temperature of 20 mK and a vector magnet system capable of applying 6,1,1 T in the $y,x,z$ directions, respectively. The differential conductance $G = \frac{dI}{dV_x}$ was measured using standard lock-in techniques, where $V_{AC}$=1.25 mV for data in Figs 1c and 3, $V_{AC}$=5 μV for data in Figs 1d and 2 and $V_{AC}$=20 μV for data in Fig. 4.

**Data availability**. The data that support the findings of this study are available from the corresponding author on reasonable request.

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

## Acknowledgements

We thank Karsten Flensberg, Jens Paaske, Guru Khalsa, Colin Clement, Andrew Higginbotham and Aharon Kapitulnik for helpful discussions. We thank Shivendra Upadhyay for technical support. This work was supported by the Villum Foundation, Lundbeck Foundation and the Danish National Research Foundation. M.L. acknowledges support from the Swedish Research Council.

## Author contributions

G.E.D.K.P. designed the experiments; G.E.D.K.P., F.T. and Y.C. fabricated the samples; G.E.D.K.P. carried out the measurements and analyzed the results with input from T.S.J.; M.L. carried out the theoretical analysis and simulations; G.P. and T.S.J. wrote the manuscript with input from all co-authors.

## Additional information

**Competing interests:** The authors declare no competing financial interests.

