## [Peer Review File · Nature Communications]

Reviewers' comments:

Reviewer #1 (Remarks to the Author):

In the manuscript titled "Gapped excitation spectrum and charge Kondo effect in a negative-U oxide quantum dot", Prawiroatmodjo et al. studied the transport characteristics of a random quantum dot at LAO/STO interface and claimed the observations of a pairing ground state and charge Kondo effect. This work is an interesting extension of Ref. 5, which reports the existence of preformed electron pairs far above superconducting conditions at the LAO/STO interface. Beyond repeating the electron pairing observations in Ref. 5, the authors provided some new transport calculations and studied temperature dependence in the weak and strong coupling regimes to support their new claims. This work should be very interesting provided the reach physics and potential applications of LAO/STO nanostructures. However, I am mainly concerned on disorder related effect and the claim of charge Kondo effect. So I would recommend to publish only if the authors can fully address my concerns. In the following, I will comment the manuscript in detail.

a) To my understanding this work is thoroughly based on 1 device, which is OK. However, more devices would certainly increase the credibility of the analysis and claims.

b) To my impression, Ref. 5 did not conclude negative U Anderson model, where pairs are localized and interact with the conducting Fermi sea. Rather it was one of the possibility. This might shake the foundation of this paper, however, I think it might work with tiny quantum dot since charges can be localized in the dot. Can the authors comment?

c) The quantum dot in the work was formed when the constriction was pinched off by the split gates. This process could produce multiple dots in principle, especially a large gate range was explored. This could explain the large, non-uniform gap in the diamonds as well, since you will need a large bias window to pass current through dots in series. At least, if a small dot is in series with the main QD, it will overstate the value of U. One way I think the authors can try is to relate the value of U to critical magnetic field. If they agree with each other, then the multiple dots picture can be downplayed.

d) What is the physical meaning of U? What is the relationship to the pairing energy? The authors should explain well to the readers.

e) In the 3rd paragraph on page 3, the authors stated "At high V_{sd} , a discrete excitation spectrum is clearly observed as lines parallel to the diamond edges confirming that the observed pairing is not due to a conventional superconducting QD."

I do not quite get the purpose of this paragraph. The discreteness of the excitation spectrum is related to the number of electrons in the dot and have nothing to do with the nature of pairing. The conventional metal superconducting SET contains orders more electrons than the LAO/STO quantum dot. I think a more reasonable comparison in terms of the nature of pairing is to look at the pairing magnetic field. I can imagine the pairs in metallic dots break up at magnetic field $< Hc_2$, opposite to the LAO/STO QD.

f) In the 6th paragraph on page 3, the authors states "At the $N=0,2$ degeneracy point, low-bias conductance at $B = 0$ results from a combination of second-order pair tunneling and thermally excited sequential tunneling" in order to justify the temperature dependence. If this is the case, should you see a triple splitting in magnetic fields? It is hard to tell due to the quality of the data.

g) In Fig. 3c, the increase of conductance with increasing temperature can also be explained by the multi-dot picture.

h) When describing the fitting in Fig.3, the authors mentioned couple of times that it agrees with calculations. For example, some description on the bottom of page 4, "The result is consistent with our calculations and confirms the discrete spectrum of the QD. For the $B = 0$ case, the combination of thermally excited single electron transport and second order pair tunneling leads to a completely different non-monotonic decrease in peak height (Fig. 3c) which is in good agreement with the experimental results." Where exactly are the calculations? There seems some simple descriptions in Sec. 5 in supplemental information, however it is too simple to justify the agreement. By the way, Eq. (5) in the supplemental information needs a reference. I suppose it is

originally from Beenakker's work (Phys. Rev. B 44, 1646 (1991)), but it is for single electron tunneling not for pair tunneling.

i) I am not convinced by the claim of charge Kondo effect. In Fig. 4, clearly the device is superconducting (e.g. the high zero-bias conductance at $V_g=1.25$ V in Fig. 4a) due to increased coupling. It is essentially a bad supercurrent transistor (Nature 439,953 (2006)). Increasing the temperature suppresses superconductivity so G decreases. Further increasing the temperature will increase G due to increased thermal excitation at high temperatures. This explains the conductance minimum in Fig. 4h.

The T_c and B_c the authors used were extracted from the outer leads at one V_g , which should be different from actual QD and inner leads. I suspect at $B_z=300$ mT the superconductivity is not fully suppressed in the QD at this gating condition, to explain the low temperature side T dependence in Fig. 4g.

j) Continuing discussion in i), the charge Kondo effect is a result of interaction of Fermi sea with localized electron pairs, so the transport should be single electron in nature. But here Fig. 4b shows pairing is still the ground state. In addition, I suppose the Kondo temperature extracted (400 mK) should match the pairing temperature (~ 1 K extracted from the pairing critical field 1 T in Fig. 4b) if it is the charge Kondo effect.

k) The authors should check the temperature dependence of an open quantum dot in the normal state, i.e. the wire without getting pinched off. Can you see a resistance minimum like Ref. 39 in TI doped PbTe?

Some other minor comments

l) Typo in line 15, page 3. "completed"->"depleted"

m) Units in X axes of Fig. 4g,h are wrong.

n) $N=0,2$ is misleading since the dot can not be fully depleted.

o) It would be helpful to include more transport calculation details in Sec. 4 in the supplemental information. Several simple generic Hamiltonians can not directly lead to the simulations in Fig. 2e.

I am sorry for so many comments. I think it is an interesting paper and I like it in some ways. However, scientifically I think this paper needs to be more rigorous and aim at a different perspective possibly.

Reviewer #2 (Remarks to the Author):

The authors report transport measurements in gate-defined quantum dots produced in the pinched-off channel of a quantum point contact in LAO/STO heterojunction 2d electron gas. The 2deg in the leads is superconducting below about .25K unless an external out-of-plane magnetic field is applied. In the more pinched-off/weaker-coupling limit, they find Coulomb blockade transport that is well described by a model incorporating a negative on-site interaction energy U , such that, at zero transverse magnetic field, the occupancy of the localized dot site goes up by 2 at each blockade crossing - in other words, the dot hosts localized pairs of electrons. Application of a transverse B_y Zeeman splits the paired occupancy states such that single-charge tunneling is restored. In the strong coupling limit, the authors identify resonances that have systematic variations consistent with expectations of the charge Kondo effect.

The experimental results are very interesting and clear, and the agreement with theoretical

expectations (e.g., Fig. 1d,e; Fig. 2) is impressive. The strong coupling regime resonances at zero bias that grow with decreasing temperature do seem suggestive of the charge Kondo effect. I support publication after minor revisions. Specific comments:

1) The authors emphasize (Fig 1) data for one particular resonance and one particular cooldown. They find a negative U with a magnitude of 160 μeV . On subsequent cooldowns, or for other resonances, do they find variation of this magnitude of U ? Have they seen these transport signatures in other devices fabricated from different growths? This information could help pinpoint the mechanism behind the negative U . Moreover, a weaker negative U would allow access to the Zeeman splitting regime within the perpendicular field range of their vector magnet. A comparison between in-plane and perp field Zeeman effects would be instructive. Basically, I'm trying to get a sense of how reproducible the observations are, and whether this device is unusual or typical. Again, this could go a long way toward clarifying the source of effective attractive interactions.

2) In the nominal strong coupling regime, many more resonances are seen (Fig. 4a). The argument is that these correspond to other dots in the "leads" of the primary dot. Naively I'd think that some of these (at smaller positive gate voltages in particular) should be very similar to the weak coupling case examined in Fig. 1, 2. Is this correct? Do they have similar negative U response? For the particular resonances examined in Fig. 4c-h, and S9&10, how were those resonances chosen? Are those the ones that happen to show increasing conductance as T is lowered? Given the proliferation of resonances (and effective dots), a more extensive description of how particular states are identified and selected for analysis would be helpful.

3) I suggest that the authors look at the literature regarding the charge Kondo effect from Cornaglia et al (e.g., [dx.doi.org/10.1103/PhysRevB.71.075320](https://doi.org/10.1103/PhysRevB.71.075320); [dx.doi.org/10.1103/PhysRevLett.93.147201](https://doi.org/10.1103/PhysRevLett.93.147201)). While those theorists consider a particular implementation of the charge Kondo effect that is not this system, they have theoretical models that examine a range of effective U that is of interest.

Reviewer #3 (Remarks to the Author):

The authors present very interesting results on physics on confined structures on LAO-STO interfaces. The interplay of magnetism and superconductivity in these materials have opened many questions - which still remain open, and the present manuscript may give indications on some key issues from an unconventional new perspective. As far as general interest is concerned, the present paper perfectly qualifies for Nature Communications. After having carefully reviewed the paper also from the technical sides, I reached the conclusion that it provides strong and solid evidence for the nontrivial interpretation based on emergent attraction. It also provides a first possible experimental realization of a negative U Anderson model. Therefore I recommended this paper for publication under the condition that the remarks and questions below will be fully addressed.

Evidence is given for the existence of a high energy doublet. The latter is interpreted using a negative U attractive model. This main physical picture actually is not new but rather a confirmation of the results from the J. Levy group, published in Nature in 2015. This is not a criticism. In my view the present paper brings a very nice and important confirmation of these nontrivial claims, in a different quantum dot, which strongly suggests that the negative U picture is a correct interpretation and appears as a generic feature of the parent material. The authors should explain more clearly that this main line of interpretation, which is a significant portion of the paper, is indeed a confirmation of the non-trivial claims which were already made in essence from the J. Levy group. If I missed something and the interpretation is not identical then a proper comparison should be given.

Regarding the formation of the quantum dot at a quantum point contact, which is another surprising and unexpected situation, this paper is again in close contact with another very recent paper, not cited, by Maniv et al. Phys. Rev. B 94, 045120 (2016). The authors should clearly state that the formation of a quantum dot in their experiment is supported by this recent other experiment of Maniv et. al.

Can the authors estimate the size if the quantum dot?

The authors show as the main figure of the paper, the conductance versus gate voltage at various values of magnetic field, but keeping always a finite magnetic field - to destroy superconductivity as stated in the paper. The authors do not show the zero magnetic field data, however. Why? This important piece of information should be shown as an additional figure (or in one of the figures) at least in the supplementary material.

Then the authors turn to more detailed transport measurements and comparison with negative U Anderson predictions. Their Fig.3 shows a comparison between 2e and 1e regimes. In the 1e regime, simple sequential tunneling results predict $1/(T \cosh^2(E/2kT))$ peaks (Ref. 34). The general trend agrees with the experiment, i.e., width decreases, while height increases, at low T. However, the height does not increase as $1/T$ but seems to saturate. Similarly in Fig.3h, there is a saturation of the width, which is not explained by theory. What is the source of this saturation? does it mean that electrons do not equilibrate below 0.3K? This would put a question mark on the low T results of this experiment.

Carefully comparing the "single peak" in Fig.3a to the "double peak" in Fig.3e, few questions emerge: What happens in the right side of Fig.3a, where it seems like an additional peak starts, which the authors seem to cut? It is important to show a full picture including two consecutive Coulomb peaks, not just one in Fig.3a.

Why the 2e peak is completely un visible in Fig.1d?

Does magnetism of the LMSO hard mask have any effect on the present experiment?

Typo: in Fig.4g,h, the unit of temperature should probably be mK rather than K.

Reply to reviewers.

We thank the reviewers for their conscientious evaluation of our manuscript “Gapped excitation spectrum and charge Kondo effect in a negative-U oxide quantum dot”. Below we have answered point-by-point to all questions raised by the reviewers. The reviewer’s comments are printed in black and our reply in blue.

Reviewer #1 (Remarks to the Author):

In the manuscript titled “Gapped excitation spectrum and charge Kondo effect in a negative-U oxide quantum dot”, Prawiroatmodjo et al. studied the transport characteristics of a random quantum dot at LAO/STO interface and claimed the observations of a pairing ground state and charge Kondo effect. This work is an interesting extension of Ref. 5, which reports the existence of preformed electron pairs far above superconducting conditions at the LAO/STO interface. Beyond repeating the electron pairing observations in Ref. 5, the authors provided some new transport calculations and studied temperature dependence in the weak and strong coupling regimes to support their new claims. This work should be very interesting provided the reach physics and potential applications of LAO/STO nanostructures. However, I am mainly concerned on disorder related effect and the claim of charge Kondo effect. So I would recommend to publish only if the authors can fully address my concerns. In the following, I will comment the manuscript in detail.

We thank the referee for supporting the manuscript and providing us with very useful comments and suggestions. In the following we address each point raised by the reviewer.

a) To my understanding this work is thoroughly based on 1 device, which is OK. However, more devices would certainly increase the credibility of the analysis and claims.

The work is indeed based on results from a single device. We agree with the referee that multiple physical devices would have been preferred, however, due to the complicated process of device fabrication and the limitations of access to low-temperature equipment, further measurements were not possible. However, for consistency, the main results have been repeated for multiple cool-downs of the device, and for a considerable number of charge transitions.

b) To my impression, Ref. 5 did not conclude negative U Anderson model, where pairs are localized and interact with the conducting Fermi sea. Rather it was one of the possibility. This might shake the foundation of this paper, however, I think it might work with tiny quantum dot since charges can be localized in the dot. Can the authors comment?

It is well established that the Anderson model accurately describes the transport through quantum dots where electrons occupy discrete states localized by the confining potential, and transport occurs by tunnelling. This is well documented in conventional confined semiconductor quantum dots (QD), and our results show clearly that this model captures all essential features for the oxide quantum dots by taking into account a U negative (including the presence of the gap, the peak splittings, excitation spectrum and temperature dependencies). The agreement with the model and the discrete excitation spectrum confirms that the QD is sufficiently small that transport is essentially described by transport by a single orbital confined to the QD. In Ref. 5 on the other hand the excitation spectrum was not discussed and the ground state was modelled by a Hubbard-model with attractive interactions. In fact, for a small quantum dot where only a single orbital contributes to the transport, the Hubbard-model with attractive interactions is reduced to a negative U Anderson model (when adding tunnel couplings to the leads). We therefore believe that our model is

consistent with the interpretation in Ref. 5 in the limit that is appropriate for our system. We also want to emphasize that our interpretation are independent and does not rely on the findings in Ref. 5. The agreement between theory and experiment is clear evidence that our system is well described by a negative U Anderson model.

To make this point explicit we have added a sentence “*The model is the single-orbital version of the Hubbard model proposed in Ref. [5], tunnel-coupled to a Fermi sea and allowing calculation of the transport currents at finite bias*”

c) The quantum dot in the work was formed when the constriction was pinched off by the split gates. This process could produce multiple dots in principle, especially a large gate range was explored. This could explain the large, non-uniform gap in the diamonds as well, since you will need a large bias window to pass current through dots in series. At least, if a small dot is in series with the main QD, it will overstate the value of U. One way I think the authors can try is to relate the value of U to critical magnetic field. If they agree with each other, then the multiple dots picture can be downplayed.

We agree with the referee that a multiple-dot scenario is indeed possible when inducing QD in this way, and such behaviour is occasionally observed in gated semiconductors. However, from our previous measurement and experience of such cases in gated semiconductors both by the present authors and in the literature in general, it is clear that the diamond-patterns presented in the manuscript does not correspond to a scenario of multiple QDs in series: 1) For multi-QD devices a beating structure is observed reflecting the various QD sizes which is not observed, 2) the effective gap would not close with magnetic field in any simple way, 3) the multi-dot scenario is inconsistent with the observation that the peaks start splitting at the same magnetic field where the gap closes. Furthermore, the change in U (equivalent to the pairing field B_p) with V_g is consistent with the observation in Ref. 5. Finally, the observation that all features (including the gap) are naturally explained by the simple single-orbital model, also shows that a multi-dot scenario is not the case. We agree, however, with the referee that this point should be addressed explicitly in the manuscript and we have added a sentence “*While multiple dots could in principle be formed in the junction the device is dominated by a single quantum dot as shown below.*” (page 2).

d) What is the physical meaning of U? What is the relationship to the pairing energy? The authors should explain well to the readers.

As stated in the manuscript, U is the effective attractive binding energy; electrons experience Coulomb repulsion as well as attractive interactions of which we are not able to assign the physical origin. U is the amount by which the attractive part exceeds the repulsive and as such U is the binding energy of the pair occupying the orbital in the quantum dot. To make this more clear we have expanded the section on p.2: “*For a single energy level this is equivalent to an effective negative charging energy U [7] which favors double occupation and constitute an effective pair binding energy.*”

e) In the 3rd paragraph on page 3, the authors stated “At high V_{sd} , a discrete excitation spectrum is clearly observed as lines parallel to the diamond edges confirming that the observed pairing is not due to a conventional superconducting QD.” I do not quite get the purpose of this paragraph. The discreteness of the excitation spectrum is related to the number of electrons in the dot and have nothing to do with the nature of pairing. The conventional metal superconducting SET contains orders more electrons than the LAO/STO quantum dot. I think a more reasonable comparison in terms of the nature of pairing is to look at the pairing magnetic field. I can imagine the pairs in metallic dots break up at magnetic field $< H_{c2}$, opposite to the LAO/STO QD.

The sentence was not intended as a reference to the nature of the pairing, but rather stating the obvious: that the discrete excitation spectrum is unlike what is found in normal metallic superconducting quantum dots where a continuum of states are observed at high bias (but where the bifurcation of transport resonances can also be observed). We agree with the referee that the formulation can be misunderstood and we have now changed it to:” At high V_{sd} , a discrete excitation spectrum is clearly observed as lines parallel to the diamond edges, which is in contrast to a conventional superconducting QD with continuous density of states”.

f) In the 6th paragraph on page 3, the authors states “At the $N=0,2$ degeneracy point, low-bias conductance at $B = 0$ results from a combination of second-order pair tunneling and thermally excited sequential tunneling” in order to justify the temperature dependence. If this is the case, should you see a triple splitting in magnetic fields? It is hard to tell due to the quality of the data.

It is not exactly clear to us where the referee would expect a triple splitting with B nor what data the referee is referring to. Nevertheless, the combination of sequential and pair tunnel processes would not result in a triple-splitting when increasing B through B_p . When B exceeds B_p two sequential tunnelling peaks emerge equivalent to conventional QD. The 0-2 pair tunnelling process is, however, blocked after B exceeds B_p because of odd ground state occupation.

g) In Fig. 3c, the increase of conductance with increasing temperature can also be explained by the multi-dot picture.

Indeed a gapped system will show this, however, if the gap arose due to multiple dots in series, the pattern would be very different as discussed above. We wish to emphasize that the simple model suggested in the manuscript provides simultaneous agreement with the bias spectroscopy (ie the appearance of the gap and the excitation spectrum) as well as the temperature dependence both for $B=0$ and $B>B_p$. The model is similar to the one used in many occasions to explain the transport data in conventional quantum dots – the only difference here is that we use $U<0$.

h) When describing the fitting in Fig.3, the authors mentioned couple of times that it agrees with calculations. For example, some description on the bottom of page 4, “The result is consistent with our calculations and confirms the discrete spectrum of the QD. For the $B = 0$ case, the combination of thermally excited single electron transport and second order pair tunneling leads to a completely different non-monotonic decrease in peak height (Fig. 3c) which is in good agreement with the experimental results.” Where exactly are the calculations? There seems some simple descriptions in Sec. 5 in supplemental information, however it is too simple to justify the agreement. By the way, Eq. (5) in the supplemental information needs a reference. I suppose it is originally from Beenakker’s work (Phys. Rev. B 44, 1646 (1991)), but it is for single electron tunneling not for pair tunneling.

The experimental transport peaks is in Fig. 3 a,e with the corresponding simulations in panels b,f. The data and experiment is directly compared in panels c,d,h,g and we believe that the reader is already pointed in the text towards the results of the simulations: “Figures 3a,e show the temperature dependence of $G(V_g)$ for $B=0$ and $B=6T$, respectively, with the corresponding calculations presented in Fig 3b,f (Supplementary Section S5)”.

The model is explained in some detail in section 4 of the supplementary, and we have now expanded the discussion to include more details. However, since this is a relatively standard model used to explain conventional quantum dots, the reader is referred to the literature for further details. We have added explicit reference to the Supplementary Section S4,S5 in the above sentence and added the sentence “*The perturbation model has been shown to accurately describe transport in*

conventional QD [Leijnse2008, Koller2010], and the only difference here is that we take U negative.”

The lineshape of the peaks are well described by Equation 5 in the supplementary both in the sequential tunnelling regime and for pair tunnelling as shown in Koch et al., PRL 2006. The appropriate references have been added and we thank the referee for pointing this out.

i) I am not convinced by the claim of charge Kondo effect. In Fig. 4, clearly the device is superconducting (e.g. the high zero-bias conductance at $V_g=1.25$ V in Fig. 4a) due to increased coupling. It is essentially a bad supercurrent transistor (Nature 439,953 (2006)). Increasing the temperature suppresses superconductivity so G decreases. Further increasing the temperature will increase G due to increased thermal excitation at high temperatures. This explains the conductance minimum in Fig. 4h. The T_c and B_c the authors used were extracted from the outer leads at one V_g , which should be different from actual QD and inner leads. I suspect at $B_z=300$ mT the superconductivity is not fully suppressed in the QD at this gating condition, to explain the low temperature side T dependence in Fig. 4g.

With regards to the referees comment (i) above, we agree with the referees summary of the $B=0$ case in Figure 4a – it is indeed in complete agreement with the description and explanation given in the manuscript.

While we have carefully checked for superconductivity in the leads by separate four-terminal characterization, we cannot rule out, that even at $B_z=300$ mT, other parts of the device outside the range of the voltage probes remain superconducting as suggested by the referee. However, we find this scenario very unlikely, as this would mean that the critical field of our samples significantly exceeds the critical magnetic field of the leads and of all reported studies of superconductivity in LAO/STO [eg. Reyren et al., Science 31,1196 (2007), Herranz et al., Nat. Communication, 6, 6028 (2015)]; even for the nanoscale structures studied [eg Cheng et al., Nature, 196, 521 (2015), Cheng et al., Phys. Rev. X 3, 011021 (2013), Stornaiuolo et al., Phys. Rev. B. 95, 140502 (2017)] the critical field was merely 150 mT – ie, half of what we apply to suppress superconductivity. Finally, if parts of the sample did remained superconducting even at 300mT, the critical temperature must be significantly lower than the $B=0$ case, however, the conductance increase is observed around 300mK which is close to the conventional LAO/STO T_c values at $B=0$.

We do, however, acknowledge that we can in principle not rule out such a scenario and that this possibility should be included. Thus we have included sentence mentioning this possibility *“However, such a scenario seems unlikely, as it would require a critical temperature of 250mK at $B_z = 300$ mT while the values reported so far of the critical fields for both bulk and nanoscale LAO/STO devices do not exceed 200mT”*

We wish to emphasize that the aim of the last part of the manuscript is to point out that the qualitatively different trends observed in the strong coupling regime is consistent with the charge-Kondo model. In the weak coupling regime we show that basically all features are consistent with the predictions of a simple model which successfully describes conventional QD – only U is negative. In the strong coupling regime, new qualitative behavior is observed, but this behavior is expected for the same model with U negative. Just as spin-Kondo is inherent in the $U>0$ case, charge-Kondo is expected for $U<0$. It is thus reasonable to expect a charge-Kondo effect to emerge, and the data is consistent with this. However, in contrast to the spin-Kondo effect, the charge-Kondo effect does not come with a distinct spectroscopic feature and thus we cannot rule out that other effects could also contribute.

j) Continuing discussion in i), the charge Kondo effect is a result of interaction of Fermi sea with localized electron pairs, so the transport should be single electron in nature. But here Fig. 4b shows pairing is still the ground state. In addition, I suppose the Kondo temperature extracted (400 mK) should match the pairing temperature (~1K extracted from the pairing critical field 1 T in Fig. 4b) if it is the charge Kondo effect.

Regarding the first comment, in the QD version of the charge Kondo effect, the localized pair is provided by the orbital in the QD favoring double occupancy (effective negative U), and the leads provide the Fermi sea. Indeed the purpose of Fig. 4b is to explicitly confirm that the device remains in a regime where even ground state occupation is favored. The transport processes underlying the charge-Kondo resonance are second order co-tunneling of single electrons. This is completely analogous to the spin-Kondo case in $U > 0$ as discussed in (eg. J. Koch et al., Phys. Rev. B 75, 195402 (2007)).

The Kondo temperature T_K is determined by the coupling strength and coupling asymmetry to the leads rather than by the pairing temperature (see eg. J. Koch et al., Phys. Rev. B 75, 195402 (2007)). This is completely analogously to the spin-Kondo effect in conventional QDs, and T_K is expected to increase with coupling strength and fluctuate between charge states as the couplings depend on the details of the QD states. While the manuscript already includes the relationship between T_K and the coupling strength in the conventional Kondo problem, we agree that the manuscript does not state clearly that the same result is expected in the charge Kondo case. To accommodate this we have included now a sentence “*At the $0 \leftrightarrow 2$ charge-degeneracy point, the charge Kondo effect then lifts the transport blockade observed for weak coupling and, in agreement with the experiment, generates a conductance resonance which increases upon decreasing T below the Kondo temperature $T_K = (2U\Gamma)^{0.5}/p \exp(-\pi U/8\Gamma)$ determined by U and Γ [15]*”.

k) The authors should check the temperature dependence of an open quantum dot in the normal state, i.e. the wire without getting pinched off. Can you see a resistance minimum like Ref. 39 in Tl doped PbTe?

We did not study the behavior of the open channel. However, low-temperature logarithmic resistance upturns are often observed for bulk 2D LAO/STO samples and the literature contains several such studies where this observation has been attributed to the effect of spin-Kondo correlations which requires magnetic impurities. However, these observations may equally well be described by the charge-Kondo effect (not relying on magnetic impurities, but on negative U centers), and our manuscript – and the work of Ref.5 – clearly shows that electron pairing is a general aspect of the LAO/STO interface.

Some other minor comments

l) Typo in line 15, page 3. “completed”->”depleted”

We thank the referee for catching this – it has now been corrected.

m) Units in X axes of Fig. 4g,h are wrong.

We thank the referee for catching this – it has now been corrected.

n) $N=0,2$ is misleading since the dot can not be fully depleted.

Throughout most of the manuscript, N denotes the occupation of the orbital under investigation. To avoid confusion on this we have added a sentence “*In the following we focus on the properties of a single orbital occupied by $N = 0, 1$, or 2 electrons.*”

o) It would be helpful to include more transport calculation details in Sec. 4 in the supplemental information. Several simple generic Hamiltonians can not directly lead to the simulations in Fig. 2e. The section describing the model has been expanded as discussed above.

Reviewer #2 (Remarks to the Author):

The authors report transport measurements in gate-defined quantum dots produced in the pinched-off channel of a quantum point contact in LAO/STO heterojunction 2d electron gas. The 2deg in the leads is superconducting below about .25K unless an external out-of-plane magnetic field is applied. In the more pinched-off/weaker-coupling limit, they find Coulomb blockade transport that is well described by a model incorporating a negative on-site interaction energy U , such that, at zero transverse magnetic field, the occupancy of the localized dot site goes up by 2 at each blockade crossing - in other words, the dot hosts localized pairs of electrons. Application of a transverse By Zeeman splits the paired occupancy states such that single-charge tunneling is restored. In the strong coupling limit, the authors identify resonances that have systematic variations consistent with expectations of the charge Kondo effect.

The experimental results are very interesting and clear, and the agreement with theoretical expectations (e.g., Fig. 1d,e; Fig. 2) is impressive. The strong coupling regime resonances at zero bias that grow with decreasing temperature do seem suggestive of the charge Kondo effect. I support publication after minor revisions. Specific comments:

We thank the referee for the support and for valuable comments for improving our work. Below we answer to each point.

1) The authors emphasize (Fig 1) data for one particular resonance and one particular cooldown. They find a negative U with a magnitude of 160 microeV. On subsequent cooldowns, or for other resonances, do they find variation of this magnitude of U ? Have they seen these transport signatures in other devices fabricated from different growths? This information could help pinpoint the mechanism behind the negative U . Moreover, a weaker negative U would allow access to the Zeeman splitting regime within the perpendicular field range of their vector magnet. A comparison between in-plane and perp field Zeeman effects would be instructive. Basically, I'm trying to get a sense of how reproducible the observations are, and whether this device is unusual or typical. Again, this could go a long way toward clarifying the source of effective attractive interactions.

The manuscript is based on results from a single device. We agree with the referee that multiple physical devices would have been preferential, however, due to the complicated process of device fabrication and limitations in time and experimental capacity further measurements were not possible. The main results have, however, been repeated for multiple cool-downs of the device, and for a considerable number of charge transitions.

To show the detailed correspondence with the model the manuscript focuses on two particular resonances – one in the weak coupling and one in the strong coupling regime. Other resonances show the same behavior, however, the value of $|U|$ varies and generally become smaller as the coupling increases in agreement with the results of Ref. 5. The data is presented in the supplement, but we agree with the referee that the manuscript could benefit from explicitly stating this information and we added the following sentence

“Our results confirm the presence of effective attractive interactions and electron pairing in LAO/STO as first observed by Cheng et al [5]. While we have focused on the transport characteristics of two resonances the remaining other resonances show similar behaviour as presented in the Supplementary Information. The magnitude of U fluctuates between resonances with an overall decrease upon increasing V_g [5]. Similar characteristics were reproduced in subsequent cool-downs of the device.”

We agree that a full study of the anisotropy is a very interesting topic. However, we cannot draw any further conclusions beyond what is already presented in section S3; future studies will focus on this exact question.

2) In the nominal strong coupling regime, many more resonances are seen (Fig. 4a). The argument is that these correspond to other dots in the "leads" of the primary dot. Naively I'd think that some of these (at smaller positive gate voltages in particular) should be very similar to the weak coupling case examined in Fig. 1, 2. Is this correct? Do they have similar negative U response? For the particular resonances examined in Fig. 4c-h, and S9&10, how were those resonances chosen? Are those the ones that happen to show increasing conductance as T is lowered? Given the proliferation of resonances (and effective dots), a more extensive description of how particular states are identified and selected for analysis would be helpful.

The additional resonances in Fig. 4a correspond to successive addition of electrons to the same physical dot. We do not consider a multi-dot scenario. At higher gate-voltages the dot couplings are larger, and the dot also effectively larger leading to a smaller spacing between resonances upon sweeping V_g . To avoid confusion on this point we have added the following sentence to the manuscript: "*While multiple dots could in principle be formed in the junction the device is dominated by a single quantum dot as shown below.*"

We also want to point out that all resonances of the device, which we have studied seem to exhibit negative-U physics.

Figures S4-S13 (with the exception of the theory plots in Fig. S10) all show different examples of the same physics in different resonances. With respect to the choice of resonances we simply chose those which were most isolated and studied them in detail to avoid complications from crossing with other orbitals. With respect to the strong coupling regime, we showed four examples exhibiting the characteristic conductance increase upon lowering T. As discussed in the main text, the Kondo temperature depends on the coupling strength and coupling asymmetries and therefore is expected to vary with dot occupation (as is a well established behavior for Kondo resonances in conventional QD also). We made this more clear by adding the sentence to the text accompanying Fig. S13.

"Not all resonances in the strong coupling regime show the conductance increase at low temperature. Interpreting the upturn as a consequence of the charge-Kondo model, such behavior can be expected as the Kondo temperature depends on the coupling strength which naturally varies between resonances."

3) I suggest that the authors look at the literature regarding the charge Kondo effect from Cornaglia et al (e.g., [dx.doi.org/10.1103/PhysRevB.71.075320](https://doi.org/10.1103/PhysRevB.71.075320); [dx.doi.org/10.1103/PhysRevLett.93.147201](https://doi.org/10.1103/PhysRevLett.93.147201)). While those theorists consider a particular implementation of the charge Kondo effect that is not this system, they have theoretical models that examine a range of effective U that is of interest.

We thank the referee for pointing us towards these interesting works. As the aim of Fig. 4 of the manuscript was to point out that the qualitative trends are consistent with the charge Kondo model in the strong coupling regime and detailed fitting to theory is not attempted. Future detailed measurements are planned to facilitate accurate fitting to these models.

Reviewer #3 (Remarks to the Author):

The authors present very interesting results on physics on confined structures on LAO-STO interfaces. The interplay of magnetism and superconductivity in these materials have opened many questions - which still remain open, and the present manuscript may give indications on some key issues from an unconventional new perspective. As far as general interest is concerned, the present paper perfectly qualifies for Nature Communications. After having carefully reviewed the paper also from the technical sides, I reached the conclusion that it provides strong and solid evidence for the nontrivial interpretation based on emergent attraction. It also provides a first possible experimental realization of a negative U Anderson model. Therefore I recommended this paper for publication under the condition that the remarks and questions below will be fully addressed.

Evidence is given for the existence of a high energy doublet. The latter is interpreted using a negative U attractive model. This main physical picture actually is not new but rather a confirmation of the results from the J. Levy group, published in Nature in 2015. This is not a criticism. In my view the present paper brings a very nice and important confirmation of these nontrivial claims, in a different quantum dot, which strongly suggests that the negative U picture is a correct interpretation and appears as a generic feature of the parent material. **The authors should explain more clearly that this main line of interpretation, which is a significant portion of the paper, is indeed a confirmation of the non-trivial claims which were already made in essence from the J. Levy group. If I missed something and the interpretation is not identical then a proper comparison should be given.**

We have added two extra sentences and an extra reference to make the relation to the work from the Levy group more clear. *“We modeled the system as a single-orbital Anderson model with an effective negative U [7]. The model is the single-orbital version of the Hubbard model proposed in Ref. [5], tunnel-coupled to a Fermi sea and allowing calculation of the transport currents at finite bias.”* And *“our results demonstrate that local pair formation is a general property of LAO/STO devices and hence not unique to devices created by the AFM sketching method[5]. Further our results support the validity of the physical interpretation first presented by Cheng et al. [5]”*

Regarding the formation of the quantum dot at a quantum point contact, which is another surprising and unexpected situation, this paper is again in close contact with another very recent paper, not cited, by Maniv et al. Phys. Rev. B 94, 045120 (2016). The authors should clearly state that the formation of a quantum dot in their experiment is supported by this recent other experiment of Maniv et. al.

We thank the referee to pointing this out and we add the suggested reference to our manuscript.

Can the authors estimate the size of the quantum dot?

Unfortunately our lithographically defined geometry does not allow a sensible estimate of the effective dot size. Also the transport spectroscopy does not provide independent access to charging energies nor level spacings, which could under some assumptions lead to an estimate of effective sizes.

The authors show as the main figure of the paper, the conductance versus gate voltage at various values of magnetic field, but keeping always a finite magnetic field - to destroy superconductivity as stated in the paper. The authors do not show the zero magnetic field data, however. Why? This important piece of information should be shown as an additional figure (or in one of the figures) at least in the supplementary material.

It is unclear exactly what the referee considers the main figure of the paper. The “conductance vs. V_g at various B ” (Fig. 1d) does show the $B=0$ case also (top trace labeled $B=0$) and Fig. 2a (which is probably what the referee considers the “main figure” of the paper) is actually for $B=0$. The large-range bias-spectroscopy data for the $B=0$ case is also included in the supplementary figure S3.

Then the authors turn to more detailed transport measurements and comparison with negative U Anderson predictions. Their Fig.3 shows a comparison between $2e$ and $1e$ regimes. In the $1e$ regime, simple sequential tunneling results predict $1/(T \cosh^2(E/2kT))$ peaks (Ref. 34). The general trend agrees with the experiment, i.e., width decreases, while height increases, at low T . However, the height does not increase as $1/T$ but seems to saturate. Similarly in Fig.3h, there is a saturation of the width, which is not explained by theory. What is the source of this saturation? does it mean that electrons do not equilibrate below 0.3K? This would put a question mark on the low T results of this experiment.

From previous measurements using the same cryogenic setup, we believe that the effective electron temperature does not exceed ~ 40 mK. At the lowest temperatures the width of the conductance peaks is limited by lifetime broadening of the level, leading to a saturation, and similarly, the peak height saturates in this regime to a value determined by the coupling asymmetry of the level. We have added a sentence” *The saturation of the width at low temperature is assigned to a lifetime broadening*”

Carefully comparing the "single peak" in Fig.3a to the "double peak" in Fig.3e, few questions emerge: What happens in the right side of Fig.3a, where it seems like an additional peak starts, which the authors seem to cut? It is important to show a full picture including two consecutive Coulomb peaks, not just one in Fig.3a.

The figure showing the temperature dependence concentrates on a single resonance (just as Fig. 2 focused on the excitation spectrum of a single resonance). The “additional” peak is simply the beginning of the next transport resonance, The temperature dependence of the full gate voltage range has not been withheld from the reader and can be found in the supplemental information section figure S11.

Why the $2e$ peak is completely invisible in Fig.1d?

No peak is observed in Fig. 1d for zero bias as pair tunneling is heavily suppressed for weak coupling. In Fig. 3 we do see a small contribution at base temperature, probably because tunnel barriers have slightly increased between measurements or a more optimized setup and a slightly higher ac bias increases the signal above the noise level.

Does magnetism of the LMSO hard mask have any effect on the present experiment?

The ferromagnetism in doped manganites usually comes from the double exchange interaction in crystalline structure. The LSMO hard mask is amorphous and we do not expect it to be magnetic and no signatures of magnetism in the mask layer have been observed in any of the measurements. We agree with the referee that this information is relevant for the reader and we have added “*we note that no signatures of possible magnetism in the amorphous LSMO top layer has been observed in any of the measurements*”

Typo: in Fig.4g,h, the unit of temperature should probably be mK rather than K.
We thank the referee for catching this – it has now been corrected.

Reviewers' comments:

Reviewer #1 (Remarks to the Author):

In the revised manuscript by Prawiroatmodjo et al, I am OK with the authors response to the most of the points I raised. However, I am still not comfortable with the claim of charge Kondo effect. The authors fully agrees with my alternative explanation of Fig.4 with superconductivity related effect instead of charge Kondo. The authors responded that it was unlikely since the superconductivity at $B=300$ mT should be most likely suppressed. They enumerated a number of papers with upper critical field $B_c < 300$ mT. While it is easy to kill superconductivity in 2D due to formation of vortices. However it is likely at nanoscale. For example, following the authors' enumeration of Levy group papers, in Fig. 2b of Phys. Rev. X 6, 041042 (2016), it seems the superconductivity persists at $B=1$ T at $V_g=0$ V. So to be rigorous, I think it is appropriate to downplay the charge kondo claim, at least by removing it in the title.

Reviewer #2 (Remarks to the Author):

I am satisfied by the authors' responses (and manuscript changes in response) to my comments and those of the other referees. I am particularly gratified that the observed response repeats itself on subsequent cooldowns - this is essential if one is to place great stock in results from a single device. I recommend publication.

We are very pleased to learn that reviewers 1 and 2 are now satisfied with the manuscript.

Regarding the comment of reviewer 3, we note that Fig. 2b of Phys. Rev. X 6, 041042 (2016) does not exhibit superconductivity at $B = 1$ T as suggested by the reviewer. As stated in the paper, the critical magnetic field for their nanoscale samples is $B_c = 0.3$ T. While we find it very unlikely, we do accept the possibility of the scenario suggested by the reviewer and the manuscript contains an explicit mentioning and discussion of this:

“Further, the conductance increase for $T < 250$ mK could be explained by small regions of superconductivity remaining in the part of the leads making contact to the QD, despite the applied B_z . However, such a scenario seems unlikely, as it would require a critical temperature of ~ 250 mK at $B_z = 300$ mT. The values reported so far of the critical fields for both bulk and nanoscale LAO/STO devices do not exceed 300 mT.”

We accept the reviewer’s suggestion to remove the reference to charge-Kondo from the title and suggest instead a new title:

“Transport and excitations in a negative-U quantum dot at the LAO/STO interface”.